

# Dynamical Downscaling Data for Studying Climatic Impacts on Hydrology, Permafrost, and Ecosystems in Arctic Alaska

5    Lei Cai[1][*], Vladimir A. Alexeev[1][#], Christopher D. Arp[2],

Benjamin M. Jones[3], Anna Liljedahl[2], Anne Gädeke[2]

[1]International Arctic Research Center, University of Alaska Fairbanks, 930 Koyukuk Dr. Fairbanks, AK
99775, USA
10   [2]Water and Environment Research Center, University of Alaska Fairbanks, 306 Tanana Loop Rd.,
Fairbanks, AK 99775, USA
[3]United States Geological Survey, Alaska Science Center, 4210 University Dr. Anchorage, AK 99508-4626,
USA

15   [#]Corresponding to: Vladimir A. Alexeev (valexeev@iarc.uaf.edu)




**Abstract.** Climatic changes are most pronounced in northern high latitude regions.    Yet, there is a paucity
of observational data, both spatially and temporally, such that regional-scale dynamics are not fully captured,
limiting our ability to make reliable projections.    In this study, a group of dynamical downscaling products
were created for the period 1950 to 2100 to better understand climate change and its impacts on hydrology,

permafrost, and ecosystems at a resolution suitable for northern Alaska.    An ERA-interim reanalysis dataset
and the Community Earth System Model (CESM) served as the forcing mechanisms in this dynamical
downscaling framework, and the Weather Research & Forecast (WRF) model, embedded with an
optimization for the Arctic (Polar WRF), served as the Regional Climate Model (RCM). This downscaled
output consists of multiple climatic variables (precipitation, temperature, wind speed, dew point temperature,

and surface air pressure) for a 10 km grid spacing at three-hour intervals. The modeling products were
evaluated and calibrated using a bias-correction approach. The ERA-interim forced WRF (ERA-WRF)
produced reasonable climatic variables as a result, yielding a more closely correlated temperature field than
precipitation field when long-term monthly climatology was compared with its forcing and observational
data. A linear scaling method then further corrected the bias, based on ERA-interim monthly climatology,

and bias-corrected ERA-WRF fields were applied as a reference for calibration of both the historical and the
projected CESM forced WRF (CESM-WRF) products. Biases, such as, a cold temperature bias during
summer and a warm temperature bias during winter as well as a wet bias for annual precipitation that CESM
holds over northern Alaska persisted in CESM-WRF runs. The linear scaling of CESM-WRF eventually
produced high-resolution downscaling products for the Alaskan North Slope for hydrological and ecological

research, together with the calibrated ERA-WRF run, and its capability extends far beyond that. Other
climatic research has been proposed, including exploration of historical and projected climatic extreme events
and their possible connections to low-frequency sea-atmospheric oscillations, as well as near-surface
permafrost degradation and ice regime shifts of lakes.    These dynamically downscaled, bias corrected
climatic datasets provide improved spatial and temporal resolution data necessary for ongoing modeling

efforts in northern Alaska focused on reconstructing and projecting hydrologic changes, ecosystem processes
and responses, and permafrost thermal regimes.    The dynamical downscaling methods presented in this



study can also be used to create more suitable model input datasets for other sub-regions of the Arctic. Supplementary data are available at https://doi.org/10.1594/PANGAEA.863625.

**Key words: cryosphere, dynamical downscaling, climatic impact, bias correction, Arctic**

### 1.    Introduction

Climate change is most pronounced in high latitude regions (Johannessen et al., 2004; Serreze and Francis, 2006; Hinzman et al., 2005).   Although the exact mechanism is still under vivid discussion, Arctic amplification has been strengthening since the late 1970s, resulting in a stronger surface temperature increase than at lower latitudes, and thus a more interactive land-surface background of the Arctic (Alexeev et al., 2005; Serreze and Francis, 2006).

The physical and ecological components of the Arctic are strongly affected by the regional and global climate (Kane et al., 1991; Jorgenson et al., 2010; Grosse et al., 2011; Koven et al., 2011) and this affect is increasing rapidly (Hinzman et al., 2005; Corell 2006; Barber et al., 2008). For example, permafrost has warmed by 0.5-4 °C in the western North America Arctic since the 1970s, corresponding with air temperature increased over the same period (Smith et al. 2010, Romanovsky et al. 2010). Changes in air and ground

temperatures in this region, along with other climatic variable changes and disturbance events, have been linked to permafrost degradation and thermokarst formation (Jorgenson et al., 2006; Lantz and Kokelj, 2008; Jones et al., 2015), thermokarst lake dynamics and drainage (Plug et al., 2008; Jones et al., 2011; Jones and Arp, 2015; Lantz and Turner, 2015), thinning winter lake ice growth (Arp et al., 2012) and increases in lake open water period (Brown and Duguay, 2010), hydrologic changes (Hinzman et al., 1992; Kane, 1997; Lesack

and Marsh, 2007), and changing arctic ecosystems and habitat (Oechel et al., 1993; Hinzman et al., 2005; Tape et al., 2006; Chapin et al., 2012).   Post et al. (2009) reviewed and highlighted that many of the ecological consequences over the arctic have been underreported except for the abiotic changes, and the linkage between the climate change and the ecological consequences are still uncertain based on current studies. One big barrier from better understanding the linkage between the changing climate and the changing

cryosphere are the lack of scale of observation, both on time and space, in the Arctic.



Climate observational records are sparsely distributed in the Arctic and typically limited temporally (Vose et al., 2007). This results from sparsely-distributed observation sites, among which there are even fewer sites that record observation routinely or are well-maintained (Shulski and Wendler, 2007). Observation accuracy is also affected significantly by the harsh Arctic environment (Groisman et al., 1991; Rasmussen et

al., 2012). As a result, numerical simulation has become an obvious and powerful alternative. The newest generation of Earth System Models (ESMs) has shown their potential for Arctic climate research, through the complete coupling of each main component of the earth system, as well as global spatial coverage (de Boer et al., 2012; Mortin et al., 2013; Koenigk et al., 2014). Nevertheless, even the latest ESMs use grid spacing of around one degree—far from capable of resolving mesoscale processes like thunderstorms and

local land-air interactions, or for use in forcing hydrological models over complex terrains. Therefore, Regional Climate Model (RCM) simulations are also commonly used, in the Arctic in particular. The Weather Research & Forecast (WRF) model has been one widely-used RCM, employed for multiple research projects focusing on specific physical processes in Alaska (Mölders and Kramm, 2010; Cassano et al., 2011; Glisan and Gutowski, 2014; Mölders et al., 2014). However, there is still a lack of RCM products designed

specifically for the Alaskan North Slope and with high-resolution spatial and climatic-scale temporal coverage capable of forcing hydrological models.

Our ultimate goal is to better understand climate change and how it impacts hydrology, ecosystems, and permafrost in northern Alaska. RCM simulations represent a favorable tool for building a reasonable climatic background, not only to recapture and project the regional climate, but also to force hydrologic, ecologic, and

ground thermal regime models. This paper introduces high-resolution dynamical downscaled data sets made specifically for research applications focused on changing landscapes in northern Alaska for the late 20$^{th}$ century and early 21$^{st}$ century (Fig. 1). Firstly, the Polar Weather Research & Foreacst (Polar WRF) model dynamically downscales both reanalysis data and ESM output, producing high-resolution simulation output for further calibration. Multiple approaches of calibration including the evaluation of model output

based on comparing climatology to observation and the bias-correction using linear scaling, thus finalizing the data set specifically made for describing both the historical and projected climatic background over

northern Alaska that is capable of serving a multitude of studies and applications tailored to changes occurring in the cryosphere and the water cycle and ecosystem functioning within the Arctic system.

## 2.    Data sources

Multiple datasets are facilitated in this research, including observational data, reanalysis data, and data from the ESM.

### 2.1    Polar weather research & forecast (WRF) model

The WRF model is a flexible, state-of-the-art regional atmospheric modeling system (Skamarock et al., 2008). Since previous modeling studies using the polar MM5 model proved that regional optimization is necessary for regional climate simulation over the Arctic region (Brownich et al 2001, Cassano et al 2001), we used the polar WRF, an RCM that originated in the WRF but was upgraded based on regional modeling experiences over the polar region by the Polar Meteorology Group of the Byrd Polar and Climate Research Center at Ohio State University (Hines et al., 2009; Hines et al., 2011). This model includes new parameterization scheme settings and calibrated land-use category profiles designed specifically for modeling both terrestrial and marine component of the Arctic (Hines et al., 2009; Wilson et al., 2011, 2012). Polar WRF version 3.5.1 is used in this study.

### 2.2    ERA-interim

We chose ERA-interim as the forcing for polar WRF runs. ERA-interim is the latest generation of a reanalysis data set from the European Center for Medium-range Weather Forecast (ECMWF), serving global gridded atmospheric elements (Dee et al., 2011). As an upgraded version of ERA-40, ERA-interim has solved some of the previous data-assimilation problems that led to ERA-40 variable inaccuracy. ERA-interim represents specific progress for hydrological cycles and precipitation fields, which are key factors in determining the quality of WRF runs for this work (Dee et al., 2011). ERA-interim has shown its capability of providing generally better quality gridded climatic variables in high-latitude areas, compared to other

reanalysis products (Jakobson et al., 2012, Lindsay et al., 2014).

### 2.3   Community earth system model version 1 (CESM1)

To obtain downscaled earth system simulation products for the study of regional climate and hydrological features in the 21st century, WRF is forced by Community Earth System Model version 1 (CESM1) (Vertenstein et al., 2011; Taylor et al., 2012) as a second downscaling product. CESM1 is the latest generation of the ESM and a group member of Coupled Model Intercomparison Project Phase 5 (CMIP5) (Taylor et al., 2012). The CESM-WRF simulation is split into two periods, following CESM. "20th Century Ensemble Member #6" is employed as the forcing for the CESM-WRF historical simulation, and "RCP4.5 Ensemble Member #6" as the forcing for the CESM-WRF projected simulation. RCP4.5 (Representative Concentration Pathway 4.5) represents radiative forcing of 4.5 $Wm^{-2}$, based on the increase in $CO_2$ emissions from the beginning of 21st century. The latest global mean temperature observations prompted this choice. The observed global temperature of the first decade of the 21st century has indicated a "braking" in the accelerating global warming (Guemas et al., 2013). Other than the RCP8.5 scenario, which assembles uncontrolled greenhouse gas emissions, an observed global mean temperature increase fits more closely with the milder radiative forcing scenarios of RCP2.6 and RCP4.5, which simulate stricter greenhouse gas control policies (Moss et al., 2010; Riahi et al., 2011; Thomson et al., 2011). Although the physical mechanisms of this global warming deceleration and its potential continuity are still under debate, we elected to use a less extreme scenario as the forcing for a projected simulation in this research that will eventually cover the entire 21st century.

### 2.4   Global historical climatology network daily (GHCN-D)

Alongside ERA-interim as a forcing, Global Historical Climatology Network Daily (GHCN-D) data from National Climatic Data Center (NCDC) are chosen as the observational-based reference for the model evaluation of this research (Vose et al., 2007). Alaskan North Slope observational sites, especially those recording long-term climatological data, are sparsely distributed. Only five stations are qualified for



routinely-recorded climatic variables north of the Brooks Range in northern Alaska: Barrow, Wainwright, Deadhorse, Nuiqsut, and Umiat (Fig. 2). In this evaluation, we compare their daily precipitation (PRCP), daily maximum temperature (TMAX), and daily minimum temperature (TMIN).

### 3. Model initialization

Polar WRF downscaling simulations are conducted for the domain covering the whole North Slope region of Alaska, as well as the Brooks Range to the South and part of the Arctic Ocean to the North (Fig. 2). A ten kilometer grid spacing produces high-resolution climatic variables for northern Alaska. Temporal coverage is set at the same as their forcing—1979-2014 for ERA-WRF, 1950-2005 for CESM-WRF as the historical run, and 2006-2100 as the projected run. The starting point for all runs is the July of their first

forcing years. The first six months work as spin-up time, and are not taken into account during data analysis.

    Parameterization schemes are set to favor high-resolution, long-term runs. Multiple parameterization schemes are employed for different physical processes. For microphysical processes, the WRF single-moment 5-class scheme (WSM5) is chosen (Hong et al., 1998). The Rapid Radiation Transfer Model (RRTM) (Mlawer et al., 1997) and Dudhia scheme (Dudhia 1996) are used for longwave and shortwave radiation,

respectively. The Noah land surface scheme (Noilhan and Planton 1989) is responsible for land surface processes, and the Yonsei University scheme (Hong and Dudhia 2003) parameterizes planetary boundary layer dynamics. Simulations use the Kain-Fritsch convective parameterization (Kain 2004).

    Before finalizing all parameterization schemes, multiple one-year-long test runs were done, demonstrating that the double-moment microphysical parameterization scheme produces higher bias for

temperature and precipitation during the first half of a year than those for the other. This is due to overestimation of cloud cover, leading to a lack in simulated downward shortwave radiation at the surface. Thus, we decided to select the WSM 5 class instead, a less complicated but more mature microphysical scheme that works reasonably for long-term regional climate modeling (Hong et al., 1998). Particularly for CESM-WRF runs, as WRF does not contain the variable table of CESM data originally, we instead use WRF

intermediate files made from CESM by NCAR, available on the CISL Research Data Archive, with data set

number DS316.0 (http://rda.ucar.edu/datasets/ds316.0/).

## 4. Results

### 4.1 ERA-WRF evaluation

We evaluate model run performance by comparing forcing (ERA-interim), RCM simulation (ERA-WRF), and observations (NCDC GHCN-D) from five stations located in northern Alaska: Barrow, Deadhorse, Nuiqsut, Umiat, and Wainwright. These are the only regional stations with long-term, routinely-recorded climatic variables. We bi-linearly interpolated ERA-interim variables for the stations, and chose the nearest grid points to the stations for WRF variables. This difference in interpolation method was the result of finer grid spacing in WRF than ERA-interim. Specifically for WRF, since three out of five stations are located near the Arctic Ocean, algorithms are revised to pick the nearest points south of the stations, to ensure the chosen points are on land rather than over the ocean.

### 4.1.1 Monthly climatology

The ERA-interim, ERA-WRF, and NCDC GHCN-D datasets present similar annual precipitation long term intra-annual variation, with more rain in summer and fall than in winter and spring (Fig. 3). WRF and ERA-interim models show more precipitation than observed. Between the two data sets, ERA-interim produces a similar amount or more precipitation annually than WRF, over all five stations except for Barrow, with wet biases varying seasonally. Monthly precipitation climatology for WRF and ERA-interim during the first six months are close, while biases grow in the second half of the year. The long error bars informs the high variability of both observed and modelled precipitation, especially in summer when most heavy precipitation events happen.

Limited by the difficulties in station observations in northern Alaska, measuring precipitation, especially in winter, has long been challenging, often leading to underestimation of total precipitation, as most precipitation falls as snow instead of rain, and snow measurement can bias drastically, especially with high



wind speed and snow redistribution (Black, 1954; Liston and Sturm, 2002; Rasmussen et al., 2012). These difficulties coincide with observational winter precipitation climatology, yielding close to zero amounts for some of the stations.

On the other hand, the temperature measuring instruments has been proven trustworthy (Vose et al., 2007). Daily maximum temperature (TMAX) and minimum temperature (TMIN) are retrieved from the three-hourly ERA-WRF output and the six-hourly ERA-interim output to fit NCDC GHCN-D temperature variables. Since maximum and minimum values for temperature from ERA-interim and WRF are filtered out from daily temperature with stationary time intervals, while NCDC GHCN-D records truly daily maximum and minimum temperatures, this manner of extraction may lead to some biases during the comparison.

TMAX in ERA-interim and WRF are extracted from the temperature at 0000 UTC (3 pm local time), while TMAX in NCDC is measured as the true daily temperature maximum. WRF slightly underestimates TMAX climatologically, compared to observation (Fig. 4). This cold bias is obvious mostly during the warmest months (June to August) and the coldest months (November to February). The only exception is the Deadhorse site, at which WRF produces small warm biases (less than 3 K) from February to May. For most stations, ERA-interim also presents cold biases compared to observations, especially in the summer. In winter, however, cold biases between ERA-interim and observation are generally not as much as those between WRF and observation.

Similarly, TMIN in ERA-interim and WRF are extracted from the temperature at 1200 UTC (3:00 am local time). Unlike TMAX, TMIN monthly climatology generally shows a warm bias between ERA-interim and observation, and a cold bias between WRF and observation (Fig. 5). These biases are illustrated year round, except for March to May, when the cold bias of WRF becomes negligible for all five stations.

TMAX and TMIN jointly reflect the diurnal temperature cycle. ERA-interim is found to have less diurnal temperature variation over the North Slope. WRF, on the other hand, produces cold biases for both TMAX and TMIN during the warmest months. However, the TMAX bias of WRF in the winter is so small that it helps even the cold bias of TMIN during the same period, representing a bigger diurnal temperature variation during the coldest months. Temperature evaluation experiments by the Polar WRF group also found

a cold bias in winter and warm bias in summer on the North Slope since Polar WRF version 3.1.1 (Hines et al., 2009; Hines et al.; 2011). As found here, these biases remain in version 3.5.1. The variabilities of both TMAX and TMIN are very restricted, especially in summer when the longer period of sunlight decreases the diurnal and daily temperature variations.

**4.1.2 Statistical coherence**

Other than monthly climatology comparisons of precipitation and temperature between observations, reanalysis data, and RCM simulation, statistics further reveal an in-depth picture of RCM performance. Taylor diagrams are presented for these five stations, showing the correlation coefficients of monthly precipitation (green), TMAX (red), and TMIN (blue) climatology of ERA-interim ($\times$) and WRF (+)

compared to observational data (Fig. 6).

Both ERA-interim and WRF demonstrate monthly/seasonal precipitation and temperature variabilities. Correlation coefficients are higher than 0.7 in all cases. Among these three variables, TMAX and TMIN are more closely correlated to observation than is precipitation. Temperature coefficients are all higher than 0.95, while precipitation coefficients are in the range of 0.7 to 0.9. Regarding comparison between data sets,

however, WRF-modeled precipitation at these five stations show higher coefficients than ERA-modeled precipitation at Barrow, Wainwright, and Nuiqsut, and similar to Deadhorse and Umiat. The TMIN coefficients are also slightly higher than the TMAX coefficient, especially in comparison between WRF and observations.

Another important statistical parameter these Taylor diagrams illustrate is normalized standard deviation

(STD), representing the monthly/seasonal variability in its reference (observation). Both ERA-interim and ERA-WRF precipitation amounts have a higher standard deviation. The only exception is the STD of ERA-interim precipitation in Barrow, which is similar to observations. Regarding ERA-interim and WRF, WRF produces about 1.5 times the STD of both the ERA-interim and observation. WRF precipitation STDs are higher than those of ERA-interim in Deadhorse and Umiat, while the two differ little in Wainwright and

Nuiqsut.





Temperature STDs for both ERA-interim and WRF are close to the reference for all stations. Regarding all five stations as a whole, however, TMAX STDs mostly drop on the right side of the reference curve, while TMIN STDs drop to the left of the reference curve in general. This feature reflects the less varied temperature at night than at daytime, for both reanalysis data and the modeling product.

This statistical feature corresponds quantitatively to the panels in the last section, with higher seasonal variability in WRF monthly precipitation than in ERA-interim, especially during summer. As a result, WRF better captures seasonal precipitation fluctuation, though its climatology deviates a little further than the ERA-interim from observation. The higher resolution and favorable parameterization schemes in WRF retrieves, to some extent, the seasonal variability in precipitation over the Alaskan North Slope.

In summary, we evaluate the ERA-WRF simulation by comparing its climatology to the forcing, as well as to the observational data set. As a result, ERA-WRF has generated a reasonable regional climate for the study region in northern Alaska for the period 1980 to 2014, while greater biases are found in precipitation relative to temperature. Seasonally, biases are higher in summer and winter than in spring and fall. Before further application of this product, bias correction is necessary for this data set. Since observations over the

Alaskan North Slope are limited in density and accuracy, the ERA-interim data set instead becomes the reference in the bias correction process for this research.

### 4.2 Bias correction

In order to make this modeled data set useful to further hydrological research, bias correction is essential for climatic variables. In this case, bias correction is conducted using the linear scaling method (Lenderink

et al., 2007; Teutschbein and Seibert, 2012). A linear scaling method artificially rescales Probability Density Functions (PDFs), ensuring the corrected monthly climatology corresponds to the reference (ERA-interim, in this case). The formulas for correcting precipitation and temperature differ. The formula for precipitation:

$$P_{his}^*(d) = P_{his}(d)\frac{\mu_m(P_{ref}(d))}{\mu_m(P_{his}(d))}$$

$$P_{prj}^*(d) = P_{prj}(d)\frac{\mu_m(P_{ref}(d))}{\mu_m(P_{his}(d))}$$



in which $P^*_{his}(d)$ and $P^*_{prj}(d)$ are, respectively, daily bias-corrected historical and projected precipitation, $P_{his}(d)$ and $P_{prj}(d)$ are the originals, $P_{ref}(d)$ is the daily reference precipitation, and $\mu_m$ stands for monthly climatology.

The formula for temperature:

$$T^*_{his}(d) = T_{his}(d) + \mu_m(T_{ref}(d)) - \mu_m(T_{his}(d))$$

$$T^*_{prj}(d) = T_{prj}(d) + \mu_m(T_{ref}(d)) - \mu_m(T_{his}(d))$$

Terms in the above formulas are the same as those in precipitation correction equations.

The difference in the bias correction equations for precipitation and temperature is due to their PDFs. Climatologically, daily temperature PDF generally obeys normal distribution, with two tails, while daily precipitation PDF generally obeys two-parameter gamma distribution (Harmel et al., 2002; Hanson and Vogel, 2008). Other variables whose probability density functions behave in the same manner as temperature are bias corrected using the same algorithm. These variables include wind speed, dew point temperature, and surface air pressure. Only precipitation and temperature are considered in this bias correction evaluation, but all variables mentioned above are bias-corrected and included in the completed data set.

**4.3 CESM-WRF evaluation**

CESM-WRF acts as the second high-resolution data set, as completed for the historical period (1950-2005) and for the projected simulation (2006-2100). CESM-WRF facilitated downscaling for an earth system model describing climatic variability and controlled by its own scenarios, and thus forming a great tool for studying climate change impacts through the end of the 21st century. Like ERA-WRF, this evaluation is also necessary for CESM-WRF.

After linear scaling ERA-WRF precipitation and temperature, we use them as reference for evaluation and bias-correction of CESM-WRF. The overlapping years of ERA-WRF and the historical CESM-WRF run, 1980-2005, represent the comparison period. Firstly, Barrow and Nuiqsut are chosen as the first step of comparison, as Barrow has the best observations on the Alaskan North Slope, for its well-maintained facilities and routinely-recording observations. The site acts as a reference point, representing the climatic features of

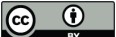



the North Slope. Further, the data quality for Nuiqsut, located just outside the northeast portion of the Fish

Creek Watershed, is critical to the reasonability and accuracy of the hydrological model forced by CESM-

WRF runs. The purpose of this evaluation is not only to validate CESM-WRF simulation, but also to produce

bias correction parameters that will be used for bias correcting the projected CESM-WRF simulation.

5        Fig. 7 shows monthly biases for precipitation and temperature at Barrow from 1980 to 2005. The left

panel shows biases before bias correction, and the right panel presents biases after applying linear scaling

bias correction. For precipitation, original CESM-WRF precipitation has a wetness bias, generally, which is

higher in summer (JJA) than in winter. After bias correction, the plot is distinctly "lighter" in color, indicating

lower biases throughout the period. Statistically, linear scaling drags mean bias down from 0.4681 to 0.0018,

and RMSE down from 0.8135 to 0.3865. Warm biases can be found in monthly mean temperature, occurring

mostly during winter. After bias correction, months with high warm biases (>8 K) decrease from the original.

Statistically, mean bias decreases from 1.5729 to 0.4357, and RMSE from 5.1694 to 4.6587.    Fig. 8 includes

the same type of plots as Fig. 7, but for the Nuiqsut station. Precipitation and temperature biases are similar

to Barrow. Linear scaling effectively corrects the wet biases for precipitation and warm biases for temperature.

What is different is that wet biases for precipitation and warm biases for temperature are higher in spring

(MMA) than are those in Barrow. Linear scaling also fixes those successfully.

        Fig. 9 and Fig. 10 contour seasonal and annual differences in precipitation and temperature between

CESM-WRF and ERA-WRF, both bias corrected. Differences between these two data sets are small:

precipitation differs less than 0.1 mm/day and temperature less than 1 K, showing the reasonability of

dynamical downscaling and the effectiveness of bias correction. CESM-WRF climatology shows a slightly

lower precipitation rate (< 0.02 mm/day) and slightly higher temperature (< 0.4 K) across the study region in

northern Alaska. Some seasonal variation features are also found for precipitation and temperature

differences.

        Although the bias is small enough for a reasonable modeled data set, seasonal variability is evident in

both the CESM-WRF and the ERA-WRF. CESM-WRF precipitation in spring (Fig. 9) and temperature in

summer (Fig. 10) exhibit opposite features from the annual difference in ERA-WRF. Also, among CESM-

WRF, precipitation over mountainous areas and inland lakes remain elevated relative to ERA-WRF. Since the configuration of these two WRF runs are identical, this small scale fluctuation in precipitation can follow only from the input field—CESM data. The cause of this heterogeneity in spatial precipitation distribution is beyond the scope of this paper, although it's interesting to witness how large-scale input causes this

heterogeneous feature for precipitation in WRF.

Linear scaling has proven effective for correcting biases but still retaining the short-term variability of the original CESM-WRF. Bias correction parameters for historical simulation are thus applied for bias correction on the projected CESM-WRF run. After this, CESM forced dynamical downscaling products for both the historical and the projected periods are completed. These data sets, as well as the reference ERA-

WRF simulation, can be applied for various research topics in climatology, hydrology, and ecology over the Alaskan North Slope, thanks to their fine grid spacing and reasonable capture of a set of climatic variables.

### 5. Discussion

This paper introduces the birth of two dynamical downscaling products forced by ERA-interim reanalysis data and CESM model output. After computational work was completed, we evaluated these

modeled variables and corrected their bias based on ERA-interim climatology and observational datasets. ERA-WRF models produce reasonable precipitation and temperature fields compared to ERA-interim. The mean precipitation amount and the seasonal variability of ERA-WRF are close to those of ERA-interim, though both of them have nearly double the annual precipitation amount relative to observational data. Temperature is, unsurprisingly, simulated better than precipitation. ERA-WRF TMAX and TMIN are

especially well-matched to observations throughout the year, although slight cold biases are found, mostly during winter, over the Alaskan North Slope, compared to ERA-interim. On the North Slope, the short and weak solar radiation in winter drag the diurnal solar radiation fluctuation down to a low level, due to the high latitude. This disappearance of variability makes solar radiation less important to driving the daily temperature cycle over the North Slope. On the other hand, cloud cover and wind advection jump out as the

important factors for surface temperature, in both summer and winter (Dai et al., 1999; Przybylak, 2000).



Winter temperature biases between ERA-WRF and its forcing are possible to dismiss by calibrating parameterizations, and seeking another scheme combination more favorable for resolving clouds and wind in northern Alaska—especially helpful when performing short-term simulations in winter months.

Bias correction is applied to all major climatic variables that are needed to drive future landscape-level

modeling efforts in northern Alaska. Bias correction is proven to have good effects in calibrating model product (Fig. 7 & 8). Previous research on CESM1 temperature modeling has found that it underestimates the seasonal cycle over the Arctic, which produces warmer winters and colder summers compared to reanalysis data does (Walston et al., 2014). CISL RDA ds316.1 applies Reynold averaging of CESM variables, based on ERA-interim that rescales 35-year climatology of CESM but maintains the perturbation term

completely (Bruyère et al., 2014; Bruyère et al., 2015). We can assume this underestimation remains in CESM-WRF, brought by its forcing. A linear scaling method for rescaling the monthly climatology/seasonal cycle is applied instead, for better bias corrections than the Reynold averaging method of both ERA-WRF and CESM-WRF. Also, it is notable that not all variables are able to be bias-corrected in this way, and the limitation results from the coarse grid of ERA-interim. For some variables that are not spatially continuous,

like the snow depth which is only over the land, the interpolation of variable field from ERA's grid to WRF's grid limits data accuracy over the coastal area, and the fact that ERA's grid does not follow the coast well makes more problematic, since ERA mistakenly recognizes some part of coastal land as part of the ocean, like Barrow. Thus, these kind of variables are not recommended to be bias-corrected before new approach of calibration is developed.

Linear scaling of CESM-WRF diminished monthly average precipitation and temperature biases, reflect the decreases of mean bias and RMSE. Taking Barrow and Nuiqsut, for example, the original CESM-WRF generally exhibits a wet bias during summer and a warm bias during winter, compared to bias-corrected ERA-WRF (Fig. 7 and Fig. 8). These are also clearly diminished by the bias correction of CESM-WRF. Precipitation correction has a relatively better effect than temperature correction, with both exhibiting virtual

biases and statistics. The majority of the climatic variables from both the ERA-WRF and CESM-WRF have been uploaded to PANGAEA after bias correction, and are available to download through the link:



https://doi.pangaea.de/10.1594/PANGAEA.863625.

Spatial variability of temperature climatology over the Alaskan North Slope has been found to be very small due to its relatively flat topography, though precipitation climatology increases from the coast to the interior because of the orographic impediment caused by the Brooks Range (Zhang et al., 1996; Serreze and

Hurst, 2000; Wendler et al., 2009). Comparison between CESM-WRF and ERA-WRF seasonal climatology coincides with this feature, showing some north-to-south gradient for temperature comparison. Precipitation comparison also yields some signals that may be relative to topography, though their existence is still uncertain, as the topographical background is offered by WRF rather than the forcing, and the land-surface backgrounds of these two runs are identical.

The linear scaling method maintains the majority of spatial distribution and temporal climate variability from the original data set, thus retaining their advantage from fine grid spacing and favorability of regional climate impact research. These two dynamical downscaling products, using the polar WRF model and forced respectively by reanalysis data and the earth system model offer major climatic variables, with high spatial resolution over our study domain in northern Alaska (Fig. 2).

**6.    Applications**

The dynamically down-scaled datasets presented in this study provide a framework for enhancing previous research efforts in Northern Alaska. For example, Scenarios Network for Alaska and Arctic Planning (SNAP) offers downscaled high-resolution daily temperature/precipitation based on CMIP3/5 GCMs (https://www.snap.uaf.edu/). The Geophysical Institute Permafrost Laboratory model (GIPL) models

soil dynamics and offers important variables on permafrost research, such as soil temperature at multiple layers, active layer thickness, freeze-up time, etc. (Marchenko et al. 2008). The GIPL, together with other two ecosystem models, comprise the Integrated Ecosystem Model (IEM), offering various ecosystem projections for all of Alaska and Northwest Canada (Rupp et al., 2015).   The previous efforts have been limited by inadequately downscaled and bias-corrected climatic datasets.   One shortcoming of the above

mentioned model-based data sets is that they are lack of a complete set of climatic variables that coupled with



components of atmosphere, land, and ocean, despite of their high resolutions. Some studies that involve surface-air interaction, such as projecting the runoff of a watershed, have to rely on multiple data sets that are independently built from each other. Some inconsistency between variables of atmosphere, land, or ocean therefore may occur. These inconsistency may lead to biases when other numerical simulations are driven by

this dataset.

The downscaled products developed by this study combine the advantages of reanalysis data set/ESM and RCM. It not only downscales the ESM's coarse grid spacing to enable regional climate studies, but also revises its lack of temperature/precipitation variabilities and extremes. Not to mention that it has gridded coverage that offsets the difficulty presented by sparse availability of observations over the Alaskan North

Slope. What makes this product outweigh others is that it offers climatic variables from multiple major components of the earth system, including the atmosphere, the land, and the ocean. All the provided variables are reasonably correlated and dependent with each other within the Polar WRF modeling framework. Thus, it is especially suitable for regional climate impacts studies that involves land-air interactions.

Our downscaled and bias-corrected product is being used to drive a grid-based Water Flow and Balance

Simulation Model (WaSiM) at watershed scales ranging from 30 to 5,000 km$^2$ in northern Alaska.   Recent droughts, such as occurred in the summer of 2007, along with uncertainty regarding hydrologic intensification (Rawlins et al., 2010) make the use of such hydrologic models valuable for understanding complex climate-permafrost-hydrology interactions (Liljedahl et al. 2016) and for simulating runoff for specific locations which lack gauging records. In the latter case, WaSiM is being applied to a small catchment in the National

Petroleum Reserve-Alaska where petroleum development is planned and baseline streamflow records are insufficiently short to evaluate any impact for changes in land-use (i.e., permanent roads, drilling pad, and lake-water extraction for operations) (Heim et al., 2014). Changes in hydrologic connectivity among rivers, streams, and lakes and how this affects fish migration and habitat use is of great interest regarding changes in land-use and even more so regionally with changing or variable climate. Its high-resolution and reasonable

precipitation, as well as other key variables, empower the WASiM recapturing the historical and projected variability spatially and temporally over a complex watershed.



Simulation of lake ice growth using temperature and snow depth data from Polar WRF is being compared to multi-temporal synthetic aperture radar (SAR) analysis of lake ice regimes, which is helping to understand changes in sub-lake permafrost, overwintering fish habitat, and availability of winter water supply for industry (Arp et al. 2012). The advantage of using our data set for this analysis is the ability to provide

continuous data for specific regions corresponding to SAR image acquisitions, whereas previous studies using station data often prove very limiting in terms of data gaps and particularly representing snow at a regional distribution. Evidence suggests that shallow lakes along the outer Arctic Coastal Plain are most sensitive to reduced ice growth (Arp et al., 2012; Surdu et al., 2014), yet this proximity is often poorly captured with coarse resolution climate datasets or station data.

Finer grid and optimized parameterization schemes of Polar WRF enable the recapturing of climatic extremes, such as that occurred in 2007 (Jones et al., 2009; Alexeev et al. 2014). Other future studies may include historical and projected extreme climatic events across the North Slope, the projected frequency and intensity of extreme climatic events under the changing climate can impact more than that from the shift of the mean, and the better capability of these downscaling products of capturing the extremes suitably facilitates

the extreme study, as well as the teleconnection of low-frequency sea-atmosphere oscillations. The climatic impact to ecosystem is difficult to estimate over the arctic since the lack of detailed the routinely observation (Post et al., 2009). These high-resolution products are able to serve as a high-quality alternative climatic background. The applications may include exploring the degrading permafrost and its deepening active layer to the releasing carbon from underground and vegetation production over the arctic tundra, and then the

impact to the habitat change of insects and large animals living based on this arctic environment.

**Acknowledgements:** Funding for this research is provided by National Science Foundation ARC-1107481 and ARC-1417300. We thank to Andrew Monaghan for converting and sharing CESM data in a WRF intermediate data format. Any use of trade, product, or firm names is for descriptive purposes only and does

not imply endorsement by the U.S. Government.



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



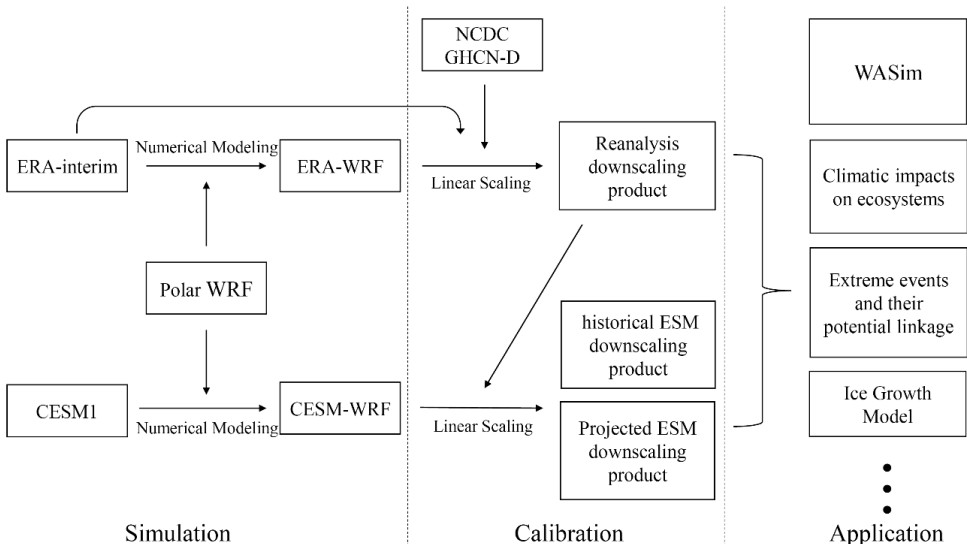

**Figure 1: The flowchart of study approach.**




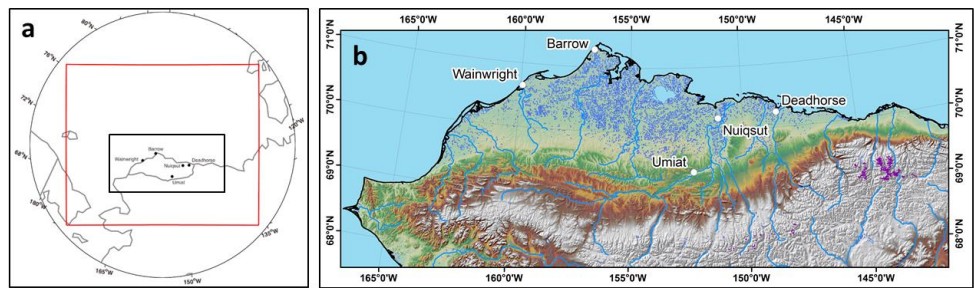

**Figure 2: The red box in figure a is the simulation domain, and figure (b) is the detailed topography of the**

**Alaskan North Slope, which is lined out as the black box in figure (a).**





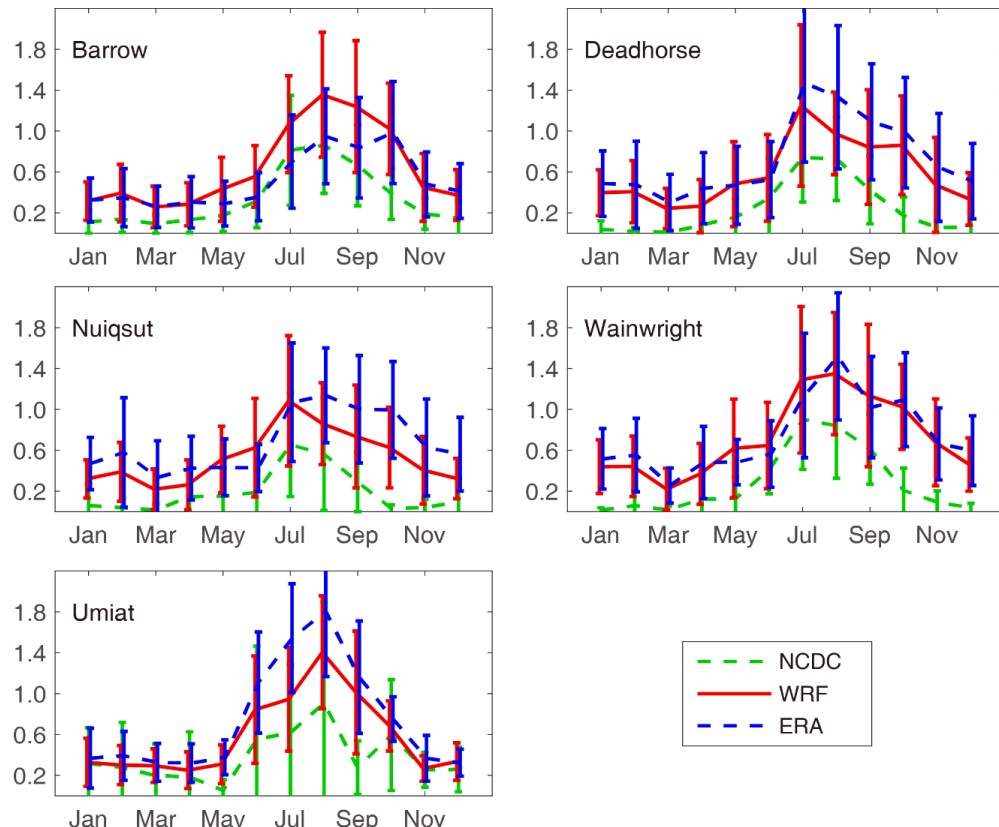

**Figure 3: Monthly mean precipitation rate (mm/day) with error bars from NCDC (green dashed line), WRF (red line), and ERA-interim (blue dashed line) at Barrow, Deadhorse, Nuiqsut, Umiat, and Wainwright stations.**

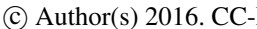
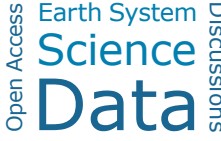


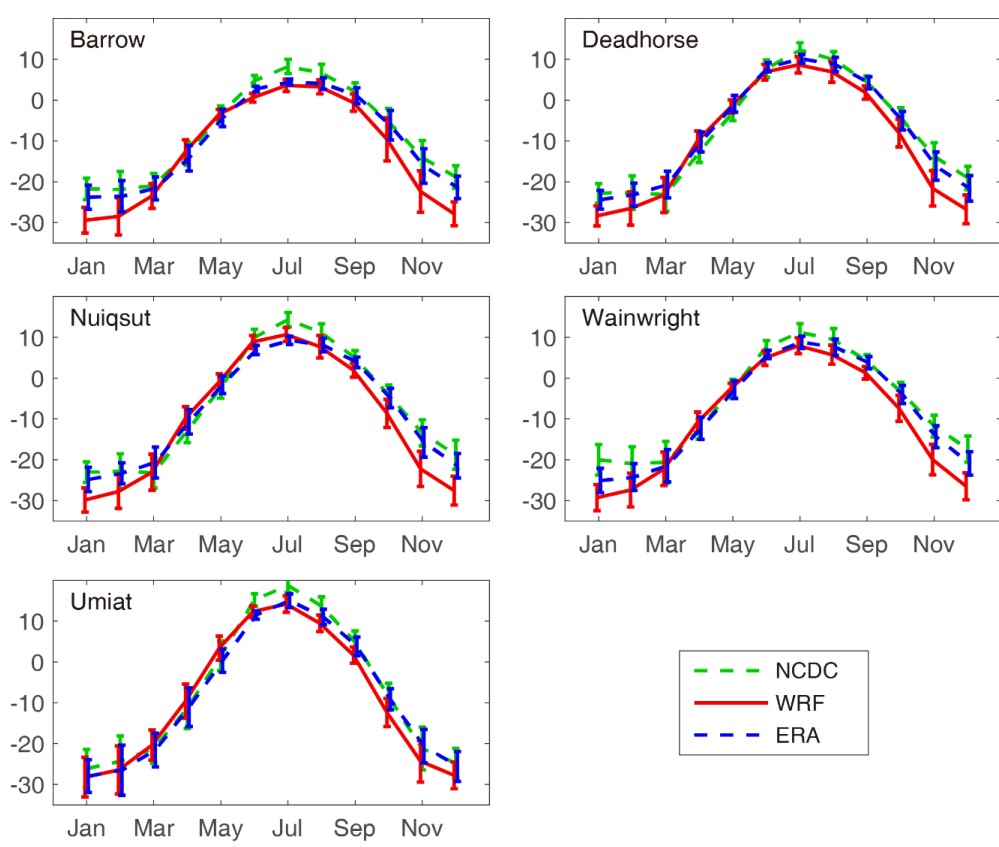

**Figure 4: Monthly mean maximum temperature (degree Celsius) with error bars from NCDC (green dashed**

**line), WRF (red line), and ERA-interim (blue dashed line) at Barrow, Deadhorse, Nuiqsut, Umiat, and**

**Wainwright stations.**



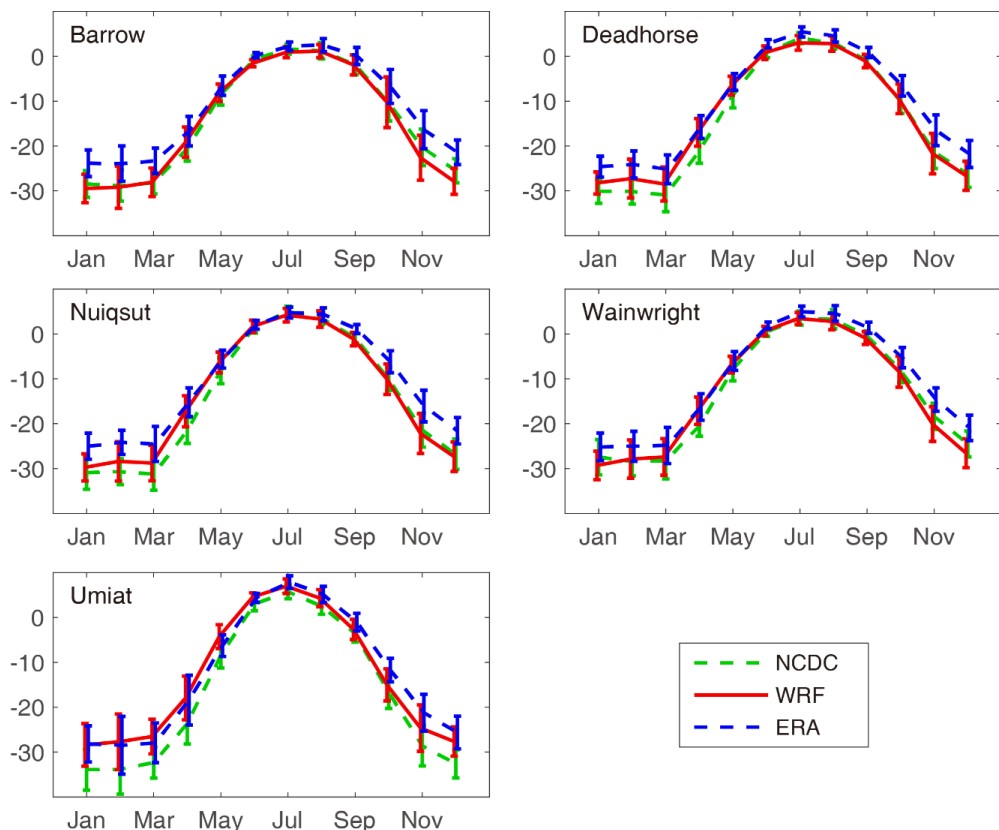

**Figure 5: Monthly mean minimum temperature (degree Celsius) with error bars from NCDC (green dashed**

**line), WRF (red line), and ERA-interim (blue dashed line) at Barrow, Deadhorse, Nuiqsut, Umiat, and**

5          **Wainwright stations.**





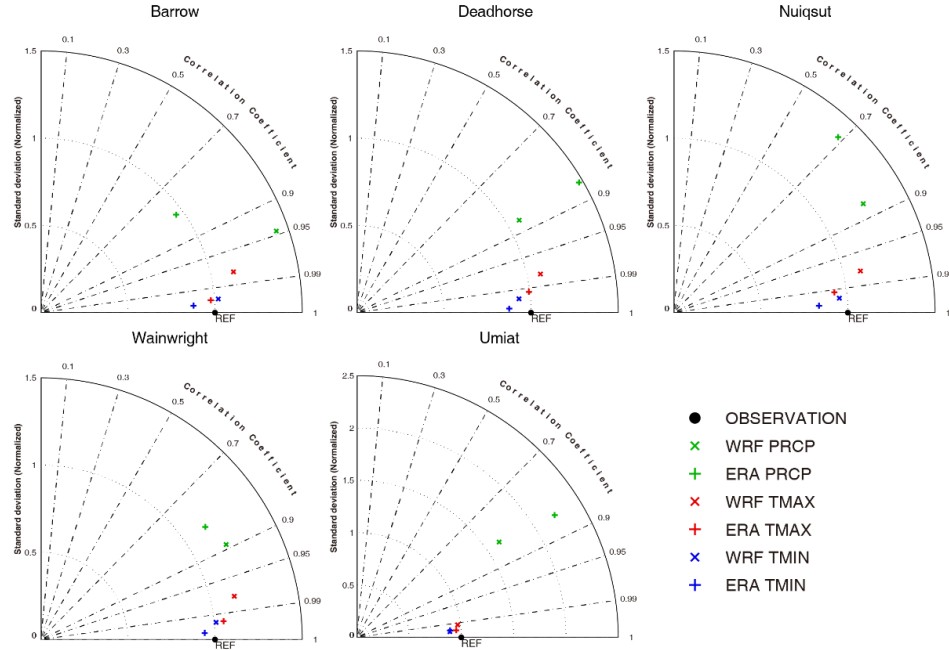

**Figure 6: Taylor diagram displaying the correlation coefficients and normalized standard deviations of**

**precipitation (green), TMAX (red), and TMIN (blue) in the WRF and ERA-interim, compared with**

**NCDC observational data (black reference dot).**




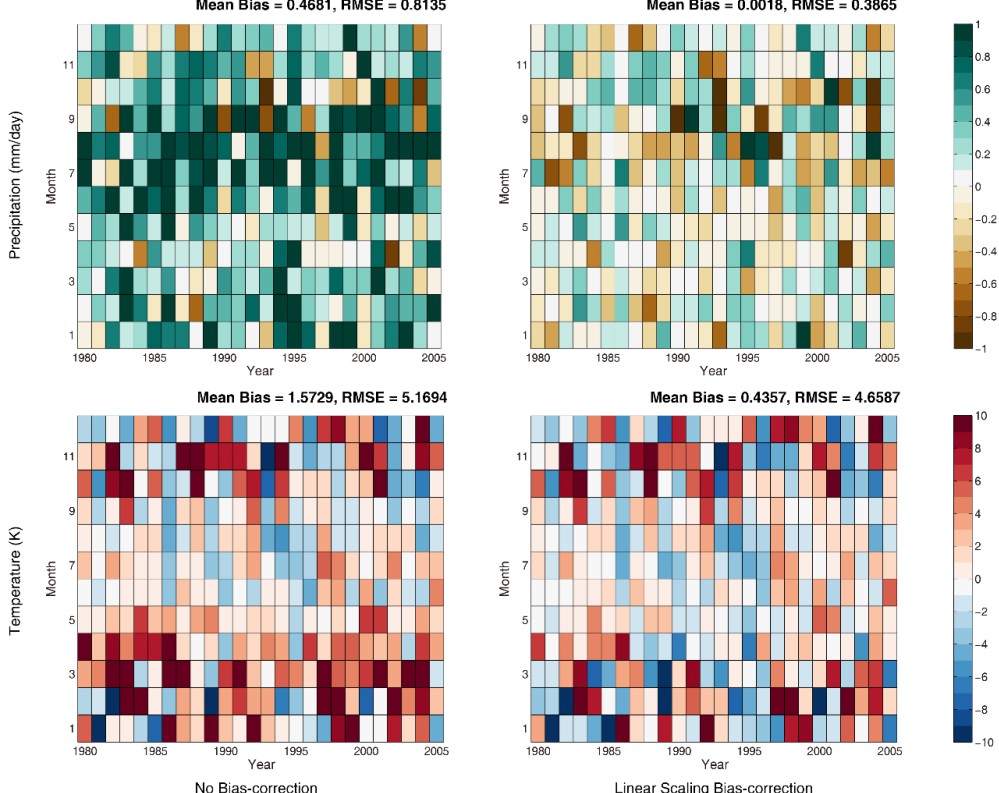

**Figure 7: The comparison between differences in raw CESM-WRF and bias-corrected ERA-WRF (left panel), as well as bias-corrected CESM-WRF and bias-corrected ERA-WRF (right panel), for precipitation rate (mm/day, upper panel) and daily mean temperature (K, lower panel) over Barrow, AK.**





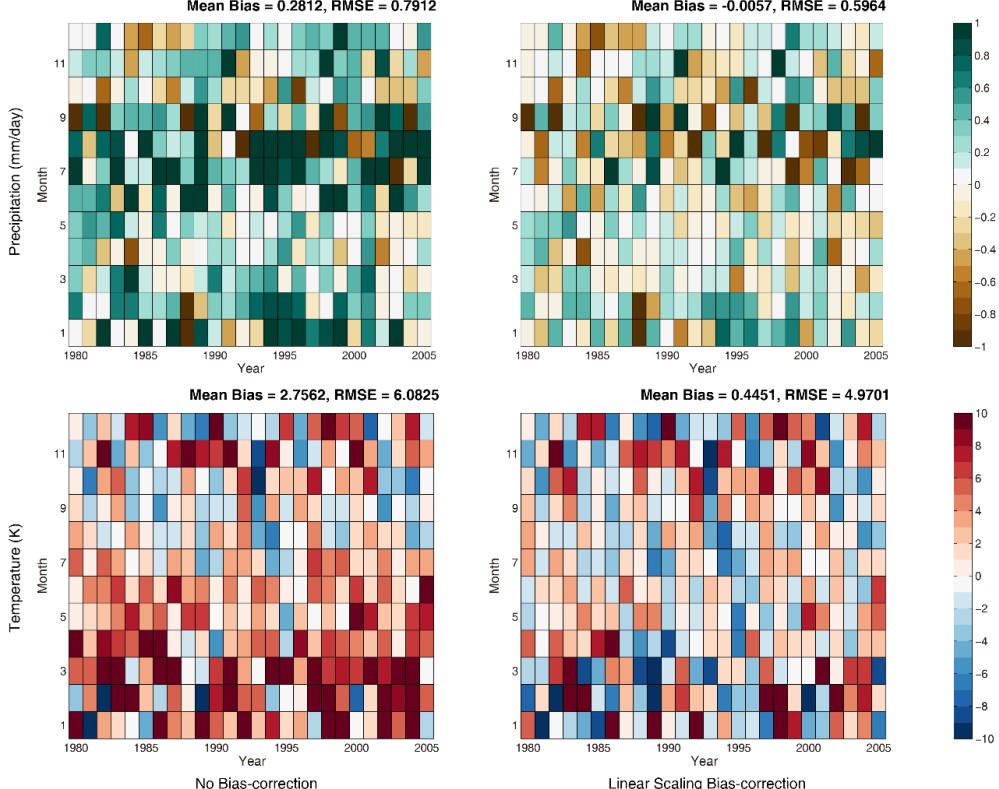

**Figure 8: The comparison between differences in raw CESM-WRF and bias-corrected ERA-WRF (left panel), as**

**well as bias-corrected CESM-WRF and bias-corrected ERA-WRF (right panel), for precipitation rate (mm/day,**

**upper panel) and daily mean temperature (K, lower panel) over Nuiqsut, AK.**




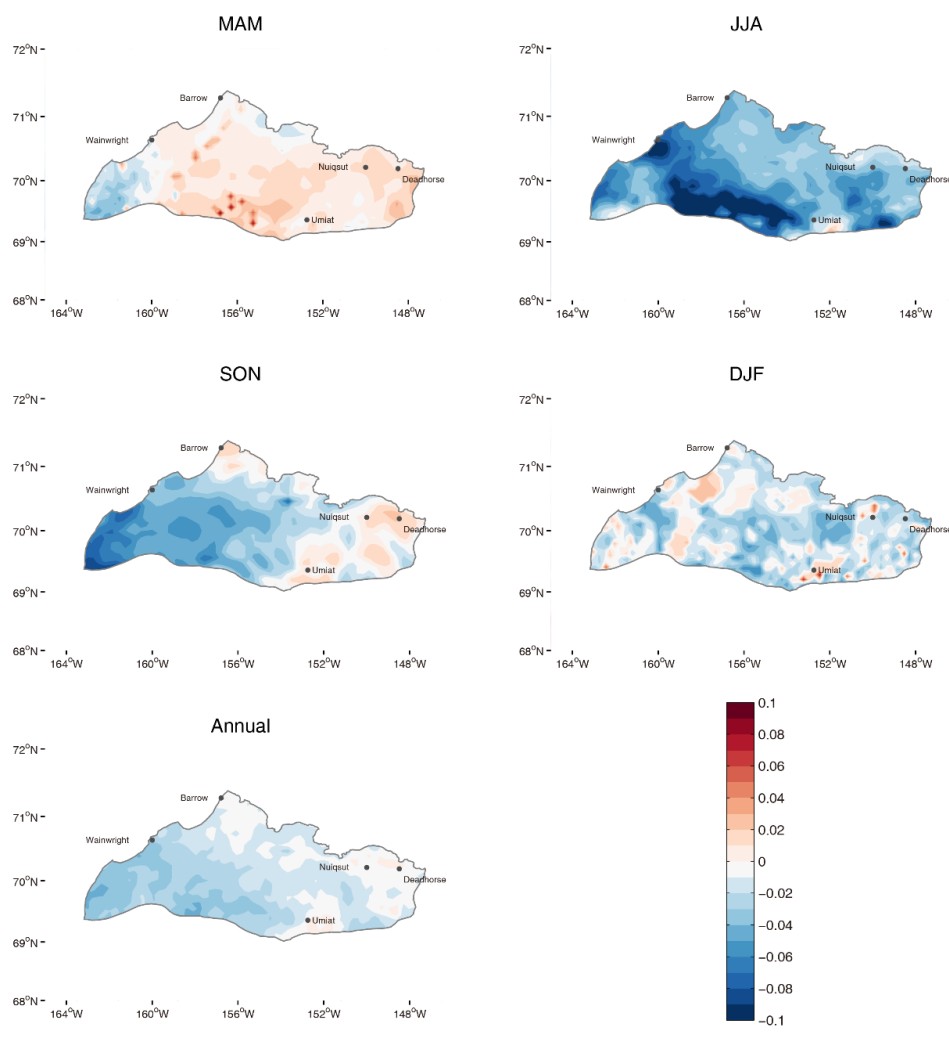

**Figure 9: Seasonal and annual differences in precipitation rate (mm/day) between CESM-WRF and ERA-WRF,**

**both bias-corrected by linear scaling method.**




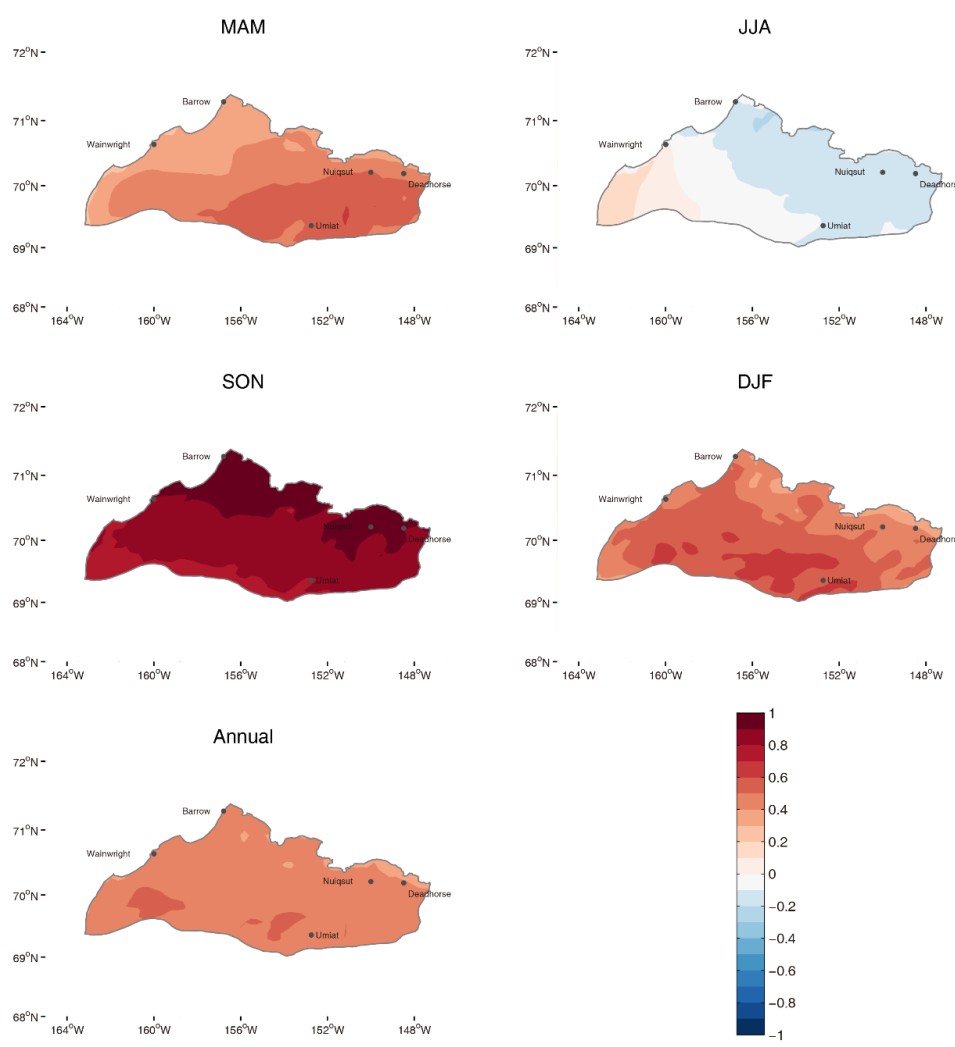

**Figure 10: Seasonal and annual differences in daily mean temperature (degree Celsius) between CESM-WRF**

**and ERA-WRF, both bias-corrected by linear scaling method.**



**Appendix: More Technical Details of developing the dynamical downscaling Data Sets**

The original WRF model source code is obtained through its official website after appropriate registration (http://www2.mmm.ucar.edu/wrf/users/), in which detailed user's guide of pre-processing, running, and post-processing the WRF is available to download. WRF V3.5.1 is compiled on pacman, a super

computing system now owned by the Geophysical Institute (GI) of University of Alaska Fairbanks (http://www.gi.alaska.edu/research-computing-systems). Then the Polar WRF upgrade provided by the Polar Meteorology Group (PMG) at Byrd Polar and Climate Research Center, The Ohio State University (http://polarmet.osu.edu/), is patched by replacing certain files in the WRF directory.

The input data sets to force WRF in this study, the ERA-interim reanalysis data, and CESM1 historical

and RCP runs, are both available on the Research Data Archive (RDA) at Computational & Informational Systems Lab, with the data set numbers of ds627.0 (http://rda.ucar.edu/datasets/ds627.0/) and ds316.0 (http://rda.ucar.edu/datasets/ds316.0/). The WRF simulation is powered by 8 16-core nodes, totally 128 cores on pacman, and takes about 18 hours for every 1-year period in the model. The whole simulation is lined up with a bunch of one-month long sub-simulations, being connected by WRF restart files. This can minimize

the loss in case the simulation crashes.

After the simulations are all done, the climatic variables that needs to force the hydrological model WASiM, including precipitation, temperature, humidity, and wind speed/direction, are extracted from the WRF output using MATLAB. Script is included as supplementary material, with the file name of wrfextract_nc.m. Specifically, the surface moisture is described by the dew point temperature, which is

calculated from the pressure, the water vapor mixing ratio, and the temperature at the surface (script is included as supplementary material, with the file name of Q2TD.m). To do the model evaluation, same variables in ERA-interim reanalysis data set are firstly interpolated to WRF's grid and then extracted. The ERA-interim variables except for precipitation are picked from the model analysis product, and precipitation is from the 3-hour forecast product. The re-gridding of ERA-interim is done by NCAR Command Language

(NCL). Documentations of NCL including functions of multiple post-processing approaches of WRF can be found on http://ncl.ucar.edu/. The re-gridding script we made is included as supplementary material, with the





file name of ERA_WRF_interp.ncl. The GHCN-daily data from NCDC can be downloaded from

https://www.ncdc.noaa.gov/oa/climate/ghcn-daily/.

After the model evaluation, the linear scaling MATLAB code for both ERA-WRF run and CESM-WRF

run are also supplemented with the file name of linear_scaling.m. Besides, all the scripts that mentioned

5   above are submitted as a .zip archive including a README file of in-detail instruction of how to use them.

There are also comments and notes in the scripts that do the same thing.