# Peer review of "Dynamical Downscaling Data for Studying Climatic Impacts on Hydrology, Permafrost, and Ecosystems in Arctic Alaska"

_Earth System Science Data, 2016_

## Referee Comment (RC1) · Anonymous Referee #1 · 1 Oct 2016

Review: Cai et al. "Dynamical Downscaling Data for Studying Climatic Impacts on Hydrology, Permafrost, and Ecosystems in Arctic Alaska"

This paper purports to present a new dynamically downscaled dataset for the North Slope of Alaska for purposes of studying climate impacts in the region. The paper presents derived historical and projected future precipitation and temperature from re-analysis (ERA-interim) and a single GCM (CESM) as well as an analysis of the biases associated with these fields.

The paper makes unsubstantiated claims about the utility of the dataset for climate impacts research, but does little to validate the methods or results and nothing to demonstrate the superiority of the approach. It analyzes nothing about hydrology, permafrost,

or ecosystems as implied in the title and therefore does not show its utility. The authors ignore the fact that the modern field of climate impacts and vulnerability assessment requires multiple climate scenarios to account for uncertainty in emissions, model construction, and climate variability. They present only one realization of 1 GCM/ESM, use rcp emissions 4.5 (not overly optimistic, but nearly), and then claim superiority to model data libraries with multiple, more finely resolved statistically downscaled GCMs and emissions scenarios without documenting the relative skill of their approach.

The methods and results are mixed up to the point it is not clear what was actually done – it sounds in the text like the ERA interim was used to drive the WRF model and then those results compared to the ERA interim and 5 stations from GHCN-D and the resulting bias correction applied to the historical and future simulations. But I have doubts about the specific methodology (see below). Most importantly, the whole point of using WRF is that all the parameters are constrained to be physically consistent- if the appropriate WRF schemes and scale are chosen (cloud microphysics, etc.) then you shouldn't be rescaling the results of just one or two of the fields because then the others aren't consistent. Additionally, if you use interpolation for the bias correction, you need to be especially careful about the scaling – it's not just regridding required but also the elevation differences and land surface parameters that lead to the superiority of the approach in the first place. This fatal flaw may be correctable by a revision, as maybe I just misunderstood the methods, but I can't evaluate it until I see what was actually done. I am deeply suspicious, however, because the paper is so poorly written and so full of conjecture that I doubt the methods are sound.

The paper fails to cite several recent papers that present peer-reviewed dynamically downscaled results for all of Alaska as well as the North Slope. I have noted these below. As such, it fails to put itself in appropriate context and actually makes false claims about its singularity and relevance.

The paper is so poorly edited that it astounds me it made it into peer-review – it should have been rejected outright on submission. I have cited many examples below, but I

[Figure]

confess I gave up after page 8 doing the diligent proof reading required. One of the authors works for the US Geological Survey - that author should make certain that a paper this poorly cited and edited never makes it outside the agency again without an FSP review. This is poor science poorly reported, and it reflects poorly on the agency for it to be sent out for review in such a state.

Comments:

Throughout: The authors go back and forth between a passive and active phrasing. Pick one, preferably active, and stick with it. It makes for shorter, more readable papers and cuts down on some of the more egregious grammatical errors.

Throughout: The authors should, if they stick with a passive voice, convert the "are" to "were" to maintain a consistent and realistic tense for the paper.

Title: It's nitpicky, but dynamical downscaling doesn't result in data. Data are measurements and observations. DD results in either model output (historical) or projections (future) – it is important in climatology to preserve the distinction, particularly in a place as data-sparse as Alaska. Also, you present this downscaling of exactly one climate model (CESM) as if it were a basis for evaluating impacts / responses, but you have no way to constrain the global-to-regional uncertainty in forcings of the WRF runs without multiple GCMs, or at least multiple realizations of the same GCM. It is not appropriate to characterize this as a basis for studying climatic impacts across regional systems. Finally, the definition of "Arctic Alaska" varies tremendously – this work is confined to the North Slope of Alaska, and the title should be amended accordingly. The US OSTP defines it more broadly. Please revise the title to accurately reflect both what is being offered and where it is pertinent to.

Abstract: "Climatic changes are most pronounced in northern high latitude regions." This is patently untrue as described. Over what time period? What variable(s)? High latitude temperature changes are larger than equatorial changes, but there are other aspects of climate and they are not universally confined to "northern" regions. What

about west Antarctica, what about places where there is no data??! This statement needs better specification before it can be true.

Abstract: "Other climatic research has been proposed, including exploration of historical and projected climatic extreme events and their possible connections to low-frequency sea-atmospheric oscillations, as well as near-surface permafrost degradation and ice regime shifts of lakes." This is not a result, and the statement as written doesn't belong in the abstract of a scientific paper. The authors can "discuss related processes, such as…..", but this is not the place to propose research. Also, "ocean-atmosphere variability" might better constrain what you mean compared to "sea-atmospheric oscillations."

P2, Line 8: "….and thus a more interactive land-surface background of the Arctic". What does that mean? Are you talking about land-surface – atmosphere interaction, and do you mean the Arctic has an intensification of such processes or that it is more "interactive" than lower latitudes? This is very poorly worded – please clarify.

P 2, Line 11: affect → effect

P2, Line 13: "0.5-4 C" If the resolution is down to 0.5 C, you should report 4.0 C.

P2 Line 14: increased → increases

P2 line 15-16: WHAT "other climatic variable changes" ? This is not very specific. This makes it sound like the permafrost degradation and thermokarst formation are the causal agents. This sentence needs to be re-written.

P2 line 18: What is "thinning winter lake ice growth"? Lake ice growth can increase or decrease (growth is a rate function) or lake ice can thin, but what does it mean as written? This is sloppy and imprecise.

P2 line 18-19: "hydrologic changes….and changing arctic ecosystems and habitat". These are hopelessly general statements. The authors need to take responsibility for more specific language here to describe what the changes are – even small changes

to these sentences would go a long way to make this sound more like a paper for a journal.

P2 line 22: "many of the ecological consequences" is not sufficiently specific.

P2 line 23: "the linkage between the climate change and the ecological consequences are still uncertain based on current studies." This sentence is confused about its plurals and singulars – linkages are, a linkage is. Please fix this.

P2 line 24: "one big barrier from better understanding. . ..are the lack of scale of observation, both on time and space, in the Arctic" This grammar is terrible. I'm 2 pages into the review and it is clear that some of the authors did not do their due diligence and read this paper before it was submitted. The number of grammar errors is inexcusable – it is irresponsible for senior researchers to assume the peer review system is the place to catch writing this raw.

P3 Line 1: Climate observational records → climate observations OR weather stations. "Limited" temporally? What defines "limited" compared to "not limited"?

P3 Lines 2-3: "This results from sparsely-distributed observation sites, among which there are even fewer sites that record observation routinely or are well-maintained (Shulski and Wendler, 2007)." Again, the authors have assumed it isn't worth their time to proof-read and appropriately edit the manuscript – they have exercised an over-developed sense of entitlement that someone else – reviewer, editor, copy editor – will edit it for them. The paper should not have been sent out for review in this state.

P3 line 4: "Observation accuracy is also affected significantly by the harsh Arctic environment." The authors need to be more specific here – do they mean the values are unobservable due to instrumentation limitations, or something else. The citations provided give clues, but not all readers will be aware of these minutiae.

P3, line 5. "As a result, numerical simulation has become an obvious and powerful alternative." Numerical simulation has little value if it doesn't assimilate actual obser-
vations – it is NOT an alternative, as this sentence implies, nor is it obvious or powerful without validation.

P3 lines 5-8: "The newest generation of Earth System Models (ESMs) has shown their potential for Arctic climate research, through the complete coupling of each main component of the earth system, as well as global spatial coverage (de Boer et al., 2012; Mortin et al., 2013; Koenigk et al., 2014)." Generation → its, not their. In addition, "complete coupling" and "global spatial coverage" do not by themselves warrant the statement "show potential" – validation, if it exists, shows potential. Please either reframe this argument or strike it. It is not logically framed on sound science as it stands.

P3 line 8-10: "Nevertheless, even the latest ESMs use grid spacing of around one degree—far from capable of resolving mesoscale processes like thunderstorms and local land-air interactions, or for use in forcing hydrological models over complex terrains." The authors appear unaware that large discrepancies in WRF runs at ∼5km vs 45km in mountainous terrain have been shown to exist elsewhere (e.g, but not limited to, Kumar et al. 2015), so complaining about the difference between 1 degree and the 10km (∼1/8th degree) runs presented here is kind of hollow.

P3 line 14-16: "However, there is still a lack of RCM products designed specifically for the Alaskan North Slope and with high-resolution spatial and climatic-scale temporal coverage capable of forcing hydrological models." Bieniek et al. (http://journals.ametsoc.org/doi/pdf/10.1175/JAMC-D-15-0153.1) was published March 2016 but you don't bother to cite that here. It provides a coherent analysis at 20km for the whole state of Alaska, including the North slope. Bieniek et al. 2015 (http://journals.ametsoc.org/doi/abs/10.1175/EI-D-15-0013.1) provides north slope WRF anomalies at 5km for purposes of assessing arctic tundra changes. The authors of this paper work in the same institution and should have known about and cited these publications. One wonders how this lack of awareness is possible.

P3 lines 18-20. "RCM simulations represent a favorable tool for building a reasonable

climatic background, not only to recapture and project the regional climate, but also to force hydrologic, ecologic, and ground thermal regime models." This is conjecture – the authors must show that the simulations are favorable and reasonable, and such a statement belongs in the discussion, not in the introduction. Alternatively, the authors could cite Lader et al. 2016 and Bieniek et al. 2015, 2016 to show this.

P3 line 20-22: T"his paper introduces high-resolution dynamical downscaled data sets made specifically for research applications focused on changing landscapes in northern Alaska for the late 20th century and early 21st century (Fig. 1)" Such products were already introduced by others. The authors need to distinguish why these are different and need to cite previous efforts that are similarly motivated.

Figure 1: Figure 1 does little to actually help the reader understand what was done here. It appears that both the ERA-interim and ERA WRF and GHCN-D were all linearly rescaled , and the resulting (how?!) reanalysis downscaling product was both used raw and also rescaled to provide both historical and projected ESM products. How many rescalings do you do before you loose the fidelity to the actual observations?

P4 line 16. Why did you choose ERA-interim? Lader et al. 2016 have shown that the different reanalyses capture different processes in Alaska, partly because of different topography. http://journals.ametsoc.org/doi/abs/10.1175/JAMC-D-15-0162.1. The ERA-interim has the second coarsest topography of the available reanalyses – why would you choose this one over the CFSR or NARR (twice finer resolution) or MERRA (finer, albeit marginally finer) resolution? You present it as an arbitrary choice, but P4-5, lines 22-1: "ERA-interim has shown its capability of providing generally better quality gridded climatic variables in high-latitude areas, compared to other reanalysis products (Jakobson et al., 2012, Lindsay et al., 2014)." The authors only show one side of this argument. See Lader et al. 2016.

P5 lines 12-19. The authors need to better defend their choice of RCP4.5, They completely fail to mention that there is a choice of RCP 6.0, and a decade of observations

is not sufficient to extrapolate which RCP we are closest to. Also, observations for 2014 and 2015 are inconsistent with the authors' citation of Guemas et al. 2013 that there is a "braking" of temperature. At this point, I am beginning to wonder if the authors are operating in a vacuum of both literature and discussion – none of this is thoroughly vetted or cited. The authors should instead have used either multiple GCMs or multiple emissions scenarios (preferably both) if the goal is characterizing future responses.

P5, line 21: The authors need to state which version of the GHCN-D they used here. In addition, NOAA requires that it be appropriately cited: "To acknowledge the specific version of the dataset used, please cite: Menne, M.J., I. Durre, B. Korze-niewski, S. McNeal, K. Thomas, X. Yin, S. Anthony, R. Ray, R.S. Vose, B.E.Gleason, and T.G. Houston, 2012: Global Historical Climatology Network - Daily (GHCN-Daily), Version 3. [indicate subset used following decimal, e.g. Version 3.12]. NOAA National Climatic Data Center. http://doi.org/10.7289/V5D21VHZ [access date]." http://www1.ncdc.noaa.gov/pub/data/ghcn/daily/readme.txt

P5, line 22: "observational-based" → observational OR observation-based, but not both.

P5-6 line 24-1: "Only five stations are qualified for routinely-recorded climatic variables north of the Brooks Range in northern Alaska." It's not that they are "not qualified" – it's that the quality of the record is insufficient given the GHCN-D standards in Menne et al. 2012 (the authors have not cited this paper). This is misrepresenting everything else as "unqualified" when in fact the reporting, length of the record, its documentation, or its QA/QC process do not meet the requirements. It's not that they're not "qualified".

P6 line 7: "A ten kilometer grid spacing produces high-resolution climatic variables for northern Alaska." That may well be true, but it doesn't belong here. In the methods, you should strike this sentence and add it in the previous one instead, "Polar WRF down-scaling simulations are were conducted AT 10km resolution for the domain covering the whole North Slope region of Alaska, as well as the Brooks Range to the South and

part of the Arctic Ocean to the North (Fig. 2)." P6, lines 11-17. The authors provide no justification whatsoever for these various choices – the citations don't indicate the relative superiority of any of them, and the authors completely ignore other researchers in the region who have made such choices – at the very least these papers should be cited, but the authors should say WHY these choices were made. Otherwise, it is arbitrary and the authors can make no claims about WHY their projections are useful. This section is confusingly written – it appears the authors did some validation (lines 18-21) and then reconfirmed their initial choice of the WSM5, but they fail to provide the results described in lines 18-20.

P7 line 6-7: "These are the only regional stations with long-term, routinely-recorded climatic variables." You already said this – it's redundant.

P7 line 7 – 8:" We bi-linearly interpolated ERA-interim variables for the stations, and chose the nearest grid points to the stations for WRF variables." If the WRF resolution is 10km, it is possible for the difference in elevation (even 150m would be about 1 C) between the station and the pixel difference to be many hundreds of meters. If you didn't account for this, your bias analysis is deeply flawed, is it not?

P7 lines 7-11 – this belongs in the methods, not the results. No results are presented here.

P7 lines 13-14: "The ERA-interim, ERA-WRF, and NCDC GHCN-D datasets present similar annual precipitation long term intra-annual variation..." this is not a grammatically correct clause.

P7 line 18: "error bars informs" → error bars indicate

P7-8: "Limited by the difficulties in station observations in northern Alaska, measuring precipitation, especiall in winter, has long been challenging, often leading to underestimation of total precipitation, as most precipitation falls as snow instead of rain, and snow measurement can bias drastically, especially with high wind speed and snow redistribution (Black, 1954; Liston and Sturm, 2002; Rasmussen et al., 2012)." This is a run-on sentence of epic proportions, with a couple of other errors embedded in it. "can bias drastically"?

Page 8: "instruments has" → instruments have

——I stopped editing sentences here——

Page 11: "Other variables whose probability density functions behave in the same manner as temperature are bias corrected using the same algorithm. These variables include wind speed, dew point temperature, and surface air pressure. Only precipitation and temperature are considered in this bias correction evaluation, but all variables mentioned above are bias-corrected and included in the completed data set." The authors need to either present analyses that show this or at least demonstrate this choice of pdf is appropriate by presenting a supplementary graphic showing they are indeed Gaussian.

"4.1.2 Statistical coherence" – Coherence has a special statistical use in both wavelet and Fourier-based time series analysis, and should not be used here as it confuses what was done here with other analyses possible for these data but not actually performed. Please choose a different title for the section – statistical comparison, perhaps.

P15 line 23-24: "The previous efforts have been limited by inadequately downscaled and bias-corrected climatic datasets." At least those are founded on a traceable, peer reviewed scientific basis rather than conjecture. The authors have failed to demonstrate such inadequacy. And most of the available climatic projections in Alaska ARE bias-corrected. Hayhoe et al., SNAP data, etc. - all are bias corrected in one fashion or other.

P16 lines 1-5: "Some studies that involve surface-air interaction, such as projecting the runoff of a watershed, have to rely on multiple data sets that are independently built from each other. Some inconsistency between variables of atmosphere, land, or ocean

therefore may occur. These inconsistency may lead to biases when other numerical simulations are driven by this dataset." These authors have not demonstrated any such "shortcomings", and while worth referencing the potential for improvement,

P16 lines 8-10: "Not to mention that it has gridded coverage that offsets the difficulty presented by sparse availability of observations over the Alaskan North Slope." This is unprofessional writing. Plenty of other datasets have gridded coverage – and the authors elected not to cite those studies.

P16 lines 10-11: "What makes this product outweigh others is that it offers climatic variables from multiple major components of the earth system, including the atmosphere, the land, and the ocean." The authors are really reaching now – each GCM has climatic variables from the atmosphere, the land, and the ocean. This paper has presented only an analysis of temperature and precipitation – no other fields. This is total conjecture and unpublishable as it stands.

The rest of page 16 is a newsletter, not a scientific paper. This is totally unsubstantiated and belongs on someone's website, not in a peer reviewed publication.

———————————————

---

## Referee Comment (RC2) · Anonymous Referee #2 · 1 Nov 2016

General comments The paper has important and valuable objectives that is, to provide a suitable dataset for the climate change impact communities in the Arctic Alaska region. Unfortunately, because of significant shortcomings in the methodology, it fails to provide a useful methodology and dataset.

One of the major issue is related to the choice of only one scenario. Taken alone, even with the best methodology, it makes the resulting dataset useless for climate studies. If this was the only issue, one could still use the methodology to generate needed multiple needed scenarios to properly evaluate uncertainties, but the remaining steps in the methodology also has its own problems that will be detailed in the specific scientific issues/comments.

For those reasons as well as the ones described in details in the following specific comments, I greatly encourage the authors to work on those issues and to re-submit the manuscript as a new one.

Specific comments

Introduction:

P3L10: Why RCMs are needed in Arctic in particular? References needed here, as RCMs are mandatory for any regional scale climate change studies

P3L15: Same comment as previous: a more detailed explanation is needed here, which RCM products are needed for Alaska and not elsewhere? Alaska Northern Slopes are not the only region where high-resolution spatial and temporal are needed for forcing hydrological models.

P3L25: Why linear scaling? The authors need to prove why one should use this specific bias correction method.

Data sources: P4L20-26: There exists other reanalysis than ERA-Interim. The choice of the reanalysis is crucial, and the authors should also explain why they did not use MERRA or other reanalyses, as the 2012 paper cited is already missing the new re-analyses available today.

P5L8-9: Only one ensemble member? Why #6? One realization of the model is not enough to take properly into account uncertainties in climate model simulations.

P5L10: The same as the previous comment for using only one RCP. The argument that the global temperature warming of the first decade is "braking" is not accountable by itself. Especially if one is interested in the second part of the 21st century, it is mandatory to use more than one RCP scenario.

P5L24-25: The author need to argument why those five stations are qualified and not the other ones.

[Figure]

P6L7: Why a 10-km grid spacing? What are the reasons of this choice? How does it impact the parameterizations?

P6L10: The authors need to argument why they chose a spin-up time of 6 months.

P6L23: Could the authors precise what "reasonably well" means in the context of the current study?

P7L7-11: Why did you choose this method for comparison? There is a need of references there, as the method to compare gridpoint data from the model and reanalysis, which represents a large surface, to station point data is crucial in the analysis being done here.

P8: Tmax and Tmin comparison methodologies can lead to large errors, as the maximum and minimum does not always occur at 3pm and 3am local time respectively. We can think of large temperature inversions during winter time, also the impacts of low cloud cover during the day and high cloud cover during the night, especially. The problem is that from the discussion it is impossible to know what are the impacts of this method of comparison on the results discussed in the followed section (statistical coherence).

Section 4.1.2: The statistical differences are stated, but no explanation is provided.

P10L15-16: This is a strong statement that needs to be proven, as it impacts the whole rest of the methodology. It is not obvious that a global reanalyses is the best dataset to use to represent precipitation at high resolution over a complex terrain.

Section 4.2: One cannot choose a bias correction methodology without doing a proper analysis of several methods. I suggest the author to, notably, take into account the paper Maraun (2013) in Journal of Climate: Bias Correction, Quantile Mapping, and Downscaling: Revisiting the Inflation Issue. What about extremes? No word is given about this aspect.

P10L23: How the author account for the non-gaussian PDF of precipitation? What

about zero values?

P11L10: Variables are bias corrected individually, thus breaking the coherence between variables. How is that taken into account in the current study?

P12L1: Basing the whole scaling bias correction parameter from only one station is not correct. This impact the whole methodology and the remaining of section 4.3.

P12L5-25: Strong biases are corrected by the method. It is not obvious that those bias correction parameters do not vary among seasons, weather circulations/types: with those large corrections more biases can be introduced if the differences between the PDFs are not the same.

P12L23: Those variation are significant, proving that the bias correction parameters should not be the same for all seasons.

Discussion and Application sections: I will not repeat the arguments stated above about the numerous significant problems in the methodology: they apply here too making the discussion to be re-assessed once the methodology has been modified appropriately.

Technical corrections No specific detailed technical corrections have been done, in light of the major problems in the methodology. A general comment about the writing is that the authors should stick to a passive OR active phrasing. Also, the Application Section is too long and should rather be included in a small discussion at the end of the paper.
* * *